DOI: 10.1038/s41467-018-04082-2　　**OPEN**

# The multiple myeloma risk allele at 5q15 lowers *ELL2* expression and increases ribosomal gene expression

Mina Ali[1], Ram Ajore[1], Anna-Karin Wihlborg[1], Abhishek Niroula [1], Bhairavi Swaminathan[1], Ellinor Johnsson[1], Owen W Stephens[2], Gareth Morgan[2], Tobias Meissner[3], Ingemar Turesson[1], Hartmut Goldschmidt[4,5], Ulf-Henrik Mellqvist[6], Urban Gullberg[1], Markus Hansson [1,7], Kari Hemminki[8,9], Hareth Nahi[10], Anders Waage[11], Niels Weinhold[2] & Björn Nilsson[1,12]

Recently, we identified *ELL2* as a susceptibility gene for multiple myeloma (MM). To understand its mechanism of action, we performed expression quantitative trait locus analysis in CD138$^+$ plasma cells from 1630 MM patients from four populations. We show that the MM risk allele lowers *ELL2* expression in these cells ($P_{combined} = 2.5 \times 10^{-27}$; $\beta_{combined} = -0.24$ SD), but not in peripheral blood or other tissues. Consistent with this, several variants representing the MM risk allele map to regulatory genomic regions, and three yield reduced transcriptional activity in plasmocytoma cell lines. One of these (rs3777189-C) co-locates with the best-supported lead variants for *ELL2* expression and MM risk, and reduces binding of MAFF/G/K family transcription factors. Moreover, further analysis reveals that the MM risk allele associates with upregulation of gene sets related to ribosome biogenesis, and knockout/knockdown and rescue experiments in plasmocytoma cell lines support a cause–effect relationship. Our results provide mechanistic insight into MM predisposition.

[1] Department of Laboratory Medicine, Hematology and Transfusion Medicine, SE 221 84 Lund, Sweden. [2] Myeloma Institute for Research and Therapy, University of Arkansas for Medical Sciences, Little Rock, AR 72205, USA. [3] Department of Molecular and Experimental Medicine, Avera Cancer Institute, Sioux Falls, SD 57105, USA. [4] Department of Internal Medicine V, University of Heidelberg, 69117 Heidelberg, Germany. [5] National Center for Tumor Diseases, Ulm 69120 Heidelberg, Germany. [6] Section of Hematology, South Elvsborg Hospital, SE 501 83 Borås, Sweden. [7] Hematology Clinic, Skåne University Hospital, SE 221 85 Lund, Sweden. [8] German Cancer Research Center, 69120 Heidelberg, Germany. [9] Center for Primary Health Care Research, Lund University, SE 205 02 Malmö, Sweden. [10] Center for Hematology and Regenerative Medicine, Karolinska Institutet, SE 171 77 Stockholm, Sweden. [11] Department of Cancer Research and Molecular Medicine, Norwegian University of Science and Technology, 7491 Trondheim, Norway. [12] Broad Institute, 7 Cambridge Center, Cambridge, MA 02142, USA. These authors contributed equally: Ram Ajore, Anna-Karin Wihlborg. Correspondence and requests for materials should be addressed to B.N. (email: bjorn.nilsson@med.lu.se)

Multiple myeloma (MM) is the second most common hematologic malignancy. It is defined by an uninhibited, clonal growth of plasma cells in the bone marrow, producing a monoclonal immunoglobulin ("M protein") that can be detected in peripheral blood[1]. Clinically, MM is characterized by bone marrow failure, lytic bone lesions, hypercalcemia, and kidney failure. It is preceded by monoclonal gammopathy of unknown significance (MGUS)[2,3], a common premalignant condition that progresses to MM at a rate of about 1% per year[4].

Several lines of evidence support that heritable factors contribute to the development of MM. Since the 1920s, several authors have reported families with multiple cases of MM and MGUS[5,6]. Systematic family studies show that first-degree relatives of patients with MM and MGUS have two to four times higher risk of MM, and a higher risk of certain other malignancies[6–11]. Recently, genome-wide association studies have identified DNA sequence variants at 18 independent loci that associate with MM risk[12–15], and show further polygenic etiology in a subset of familial MM cases[16].

One of the MM susceptibility genes is ELL2 (elongation factor for RNA polymerase II 2)[12,13] at chromosome 5q15. This gene encodes a key component of the super-elongation complex (SEC), which enhances the catalytic rate of RNA polymerase II[17,18]. ELL2 is highly expressed in normal and MM plasma cells, and helps RNA polymerase II find a promoter-proximal weak poly (A)-site in the immunoglobulin (Ig) heavy gene that is hidden in B cells, allowing Ig heavy chain messenger RNA (mRNA) to be translated to secreted Ig at a high rate[13,19,20]. Conditional B-lineage Ell2 knockout mice show curtailed humoral immune responses, reduced numbers of plasma cells, and abnormal plasma cell morphology[21–23]. The ELL2 MM risk allele is represented by ~70 sequence variants in tight linkage disequilibrium ($r^2 > 0.8$ with the first reported lead variant rs56219066[13] or the lead variant from a subsequent multi-center analysis, rs1423269; $r^2/D' = 0.96/0.98$ with rs56219066)[12]. Interestingly, the same allele that predisposes for MM also associates with lower Ig levels[13], altered Ig glycosylation[24], lower total serum protein levels[25], and an increased risk of MGUS[13], salivary gland carcinoma[26], and possibly bacterial meningitis[13].

Here we investigate the effects of the ELL2 MM risk allele. Since this allele is represented by non-coding variants (apart from one missense variant of unclear relevance[13]), we hypothesize that its effects are due to changes in ELL2 expression. Using expression quantitative locus (eQTL) analysis, we detect a negative effect of the MM risk allele on ELL2 expression in MM plasma cells. This finding is further supported by data showing that several of the risk variants map to regulatory chromosomal regions, including three that yielded reduce transcriptional activity. Interestingly, one of these (rs3777189-C) is located only 514 bp from the lead variant for ELL2 expression (rs9314162) and 2616 bp from the best-supported lead variant for MM risk (rs1423269), and diminishes binding of MAFF/G/K family transcription factors. In addition to the effect on ELL2 itself, we find that the MM risk allele perturbs the expression of genes involved in ribosome biogenesis and function.

## Results

### The MM risk allele lowers ELL2 expression in MM plasma cells.
To identify effects of the ELL2 MM risk allele on gene expression, we generated mRNA-sequencing data for CD138+ plasma cells from bone marrow samples from 185 MM patients from Sweden and Norway, and genotyped these samples for one of the linked MM risk variants at the ELL2 locus (rs3815768; Supplementary Fig. 1a). In addition, 158 of the samples were genotyped using

Illumina OmniExpress™ single-nucleotide polymorphism (SNP) microarrays, and imputed using phased haplotypes from the 1000 Genomes compendium[27].

In our mRNA sequence data, we found that the MM risk allele lowers ELL2 expression. While this effect was clearest across the distal part of the gene (exons 9–11; Pearson correlation $P = 0.007–0.01$, $\beta = -0.19$ to $-0.20$), we saw significant associations with all exons (Fig. 1a and Table 1), except with exons 7 and 8, which could not be quantified reliably for technical reasons (Supplementary Fig. 1b), and the last exon, which could not be quantified accurately because of uneven coverage in the 3′ untranslated region. Samples heterozygous and homozygous for the risk allele showed 34% and 43% lower ELL2 expression, respectively (average across exons 1–6 and 9–11) than samples homozygous for the protective allele. We also observed an allelic imbalance in expression for heterozygous individuals among rs3815768-TC heterozygotes (54.5% for T-allele vs 45.4% C-allele; $P < 0.005$). No differences in ELL2 splicing patterns were detected between the T- and C-allele using replicate multivariate analysis of transcript splicing[28].

For further validation of the observed effect, we used gene expression microarray data for CD138+ plasma cells from MM patients from Germany ($n = 658$), the United Kingdom ($n = 183$), and the USA ($n = 604$)[12,29]. In all these datasets, rs3815768-C associates with lower ELL2 expression (Fig. 1b; Fisher's inverse $\chi^2$ test combined $P = 2.5 \times 10^{-27}$ and $\beta = -0.24$ for the four datasets). Moreover, regional analysis of these data and the Swedish-Norwegian samples genotyped on SNP microarrays showed that the set of variants that most strongly influence MM risk are those that have the largest effect on ELL2 expression (Fig. 2a, b). Additionally, we observed slightly more significant P values across the second half of intron 2 and across intron 3, including both the lead variant for ELL2 expression (rs9314162) and MM risk (rs1423269). These data demonstrate a concordance between the effects of sequence variants on ELL2 expression and MM risk, and indicate that the same sequence variations at this locus affect both.

**Effect on ELL2 expression in other cell types.** While ELL2 is highly expressed in normal and malignant plasma cells, the gene is also expressed in other cell types, including red blood cell precursors, salivary gland, and pancreatic islets (Supplementary Fig. 2)[13,30,31]. Curiously, these cell types resemble plasma cells in that they produce large amounts of protein (hemoglobin, amylase, and peptide hormones), and the same allele that predisposes to MM also predisposes to salivary gland carcinoma (rs3777204; $r^2/D' = 0.96/0.98$ with rs1423269)[32]. Yet, unlike the highly reproducible effect on ELL2 expression in MM plasma cells, we could not detect any effect on ELL2 expression in mRNA-sequencing data from peripheral blood from 2515 Icelanders (Supplementary Fig. 3), nor in eQTL data from 8086 Europeans in the Blood eQTL database[33] or any of the 44 tissues represented in GTEx[34]. Although some tissues, including salivary gland, could not be studied because of lack of data, these results indicate that the effects of the MM risk allele on ELL2 expression are restricted to certain cell types.

**Identification of causal variants.** A total of 67 SNPs and 5 small insertions/deletions are highly correlated with the best-supported sentinel MM risk variant (rs1423269) and the strongest ELL2 expression variant (rs9314162) ($r^2 > 0.8$; Supplementary Tables 1 and 2). Hypothetically, some of these variants may be causal in that they alter the efficiency of ELL2 transcription, whereas others only tag the causal markers. To search for such causal variants, we considered variants

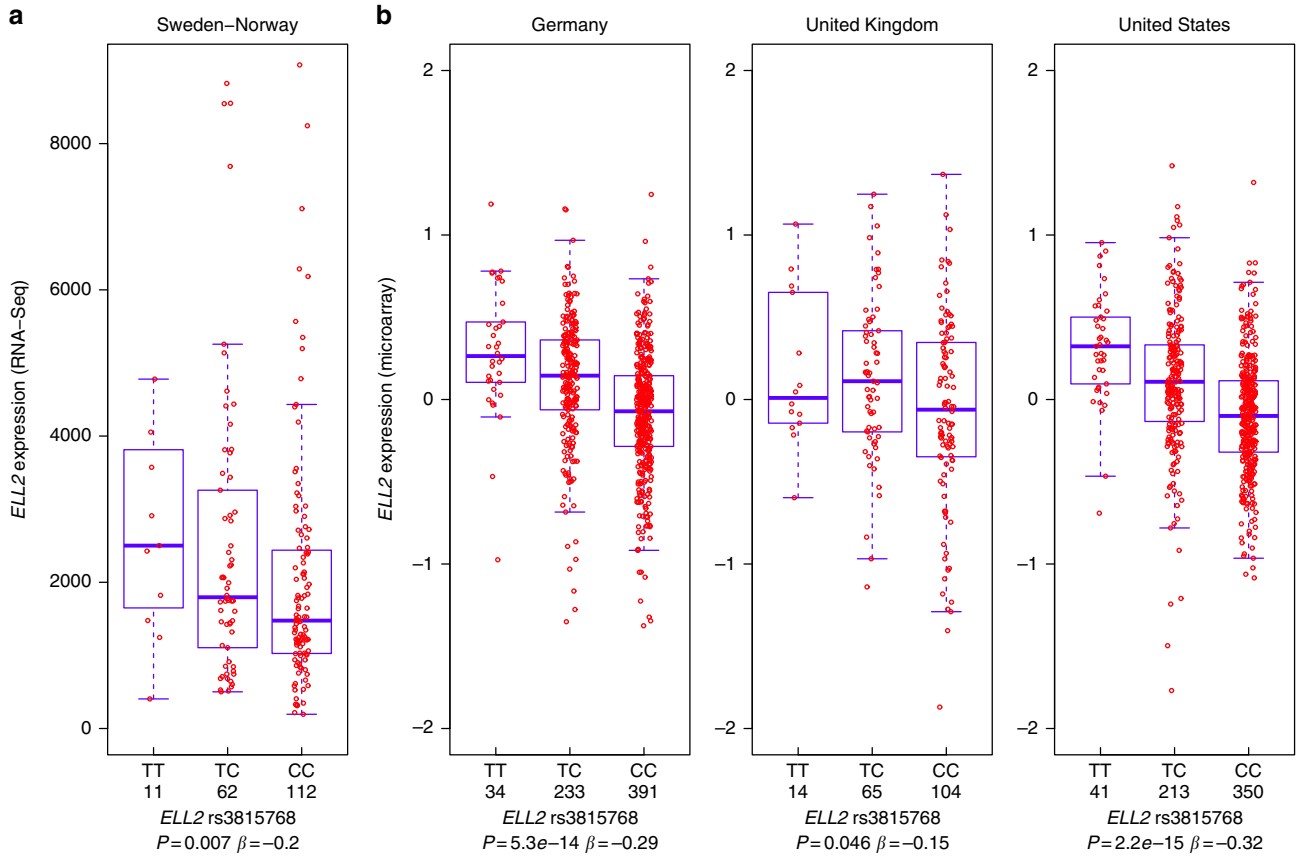

**Fig. 1** The *ELL2* MM risk allele confers lower *ELL2* expression in MM plasma cells. **a** Boxplot showing decreased expression of *ELL2* exon 10 associated with the MM risk allele (represented by rs3815768-C) compared to the protective allele (rs3815768-T) in CD138$^+$ bone marrow cells from 185 Swedish and Norwegian MM patients. Expression values are fragments per kilo base of exon per million fragments mapped (FPKM) obtained by mRNA sequencing. **b** Boxplots showing a corresponding association in Affymetrix gene expression microarray data CD138$^+$ bone marrow cells from MM patients from Germany, United Kingdom, and the US. Boxes indicate medians and the first and third quartiles. Whiskers indicate first and third quartiles plus 1.5 times the interquartile range. Outliers are plotted as individual dots. Pearson correlation $P$ values and effect size ($\beta$) indicated

| Table 1 Association testing results | | | | | |
|---|---|---|---|---|---|
| **Exon** | **Chr.** | **Position (hg38)** | **Effect size ($\beta$)** | $r^2$ | $P$ |
| 1 | 5 | 95885097–95888987 | −0.15 | 0.02 | 0.036 |
| 2 | 5 | 95889085–95889130 | −0.16 | 0.03 | 0.026 |
| 3 | 5 | 95891102–95891274 | −0.17 | 0.03 | 0.018 |
| 4 | 5 | 95895627–95895691 | −0.16 | 0.03 | 0.025 |
| 5 | 5 | 95898239–95898810 | −0.15 | 0.02 | 0.040 |
| 6 | 5 | 95900692–95900780 | −0.14 | 0.02 | 0.050 |
| 7 | 5 | 95900955–95901080 | −0.04 | 0.00 | 0.601 |
| 8 | 5 | 95906522–95906782 | −0.11 | 0.01 | 0.133 |
| 9 | 5 | 95913770–95913934 | −0.20 | 0.04 | 0.007 |
| 10 | 5 | 95919423–95919545 | −0.20 | 0.04 | 0.007 |
| 11 | 5 | 95913001–95913049 | −0.19 | 0.04 | 0.010 |
| 12 | 5 | 95911574–95912071 | −0.12 | 0.02 | 0.093 |

Association testing results for the Swedish-Norwegian mRNA-sequencing dataset ($n = 185$). Effect size ($\beta$), squared Pearson regression coefficient ($r^2$), and $P$ values indicated

in linkage disequilibrium ($r^2 > 0.8$) with rs9314162 that associate with both *ELL2* expression and MM (top-right clusters in Fig. 2b) and map to regulatory regions. To delineate regulatory regions, we used ChIP-seq (chromatin immunoprecipitation with next-generation sequencing) data for histone modifications representing enhancers and promoters, and for transcription factors, in GM12878 lymphoid cells from the ENCODE and Roadmap compendia (Supplementary Table 1)[35,36]. In addition, we generated ChIP-seq data for H3K4me3 histone marks in the L363 plasma cell leukemia cell line to delineate promoter regions relevant in plasma cells. Using our criteria, we identified eight candidate variants (rs1841010, rs9314162, rs3777189, rs3777185, rs4563648, rs6877329, rs3777184, and rs889302). All of these mapped near rs1423269 and rs9314162, and five (rs3777185, rs4563648, rs6877329, rs3777184, and rs889302) to an internal promoter in intron 2, as defined by the presence of the H3K4me3 histone mark (Fig. 2c).

To evaluate the candidate variants, we made luciferase vectors containing 120 bp of genome sequence with the respective risk and protective variants in the center (Supplementary Table 3). We transfected these vectors into three plasma cell lines (L363, OPM2, and RPMI-8226) and two cell lines representing other hematologic lineages (K562 and MOLM-13; acute myeloid leukemia cell lines with eryhtroblastic and monocytic differentiation, respectively). Consistent with our observation of an eQTL effect in MM plasma cells but not in peripheral blood, three risk variants (rs3777189-C, rs3777185-C, and rs4563648-G) yielded decreased luciferase activity relative to their corresponding protective variants in plasma cell lines, but not in non-plasma cell lines (Fig. 3a). Interestingly, rs3777189 is located only 514 bp from rs9314162; and rs3777185 and rs4563648 in the internal promoter in intron 2.

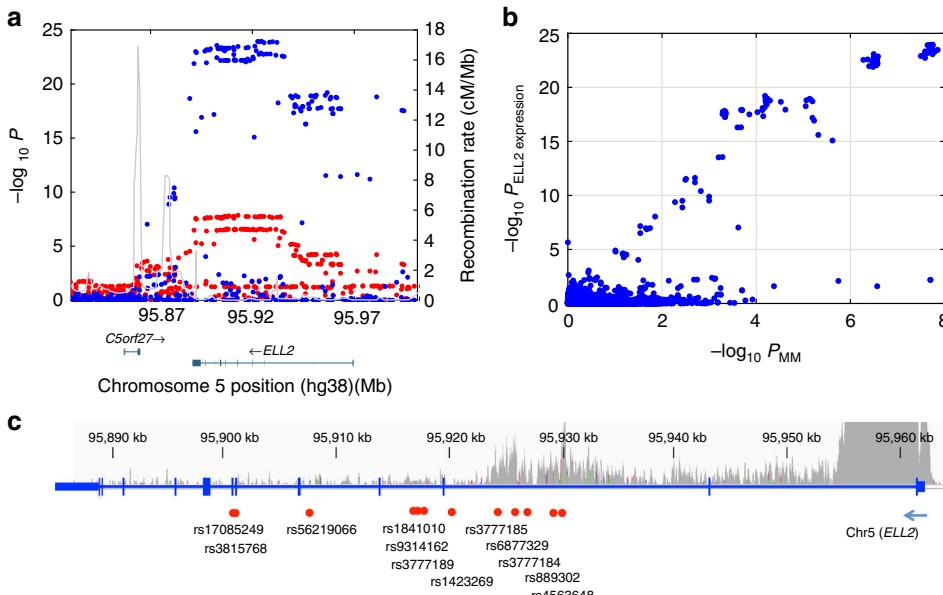

**Fig. 2** *ELL2* MM risk variants coincide with *ELL2* eQTLs. **a** Regional association plot of the *ELL2* locus at chromosome 5q15. Blue dots indicate association with *ELL2* expression based on the four sample sets (−log$_{10}$-transformed Fisher's inverse $\chi^2$ P values; meta-analysis of 158 Swedish-Norwegian and 1445 SNP microarray-genotyped samples from Germany, UK, and US). Red dots indicate association with MM (−log$_{10}$-transformed logistic regression P values from our previous Swedish-Norwegian-Iceland MM association study). The lead variant for the effect on *ELL2* expression is rs9314162. Meiotic recombination rates calculated from the 1000 Genomes compendium indicted by the gray curve. **b** Two-dimensional plot of the same association P values. The two top-right clusters contained 66 variants influencing both MM risk and *ELL2* expression. **c** Schematic representation of *ELL2*. The indicated variants are the lead variants for *ELL2* expression (rs9314162), the first reported MM lead variant (rs56219066), the best-supported MM lead variant (rs1423269), the eight variants selected for functional evaluation (rs1841010, rs9314162, rs3777189, rs3777185, rs6877329, rs3777184, rs889302, and rs4563648), and the coding variant rs3815768 used for genotyping in the mRNA-sequencing data. Gray curve indicates ChIP-seq read density for the H3K4me3 histone mark in L363 cells, and main (high peak around exon 1) and internal promoters (lower peaks across introns 1 and 2)

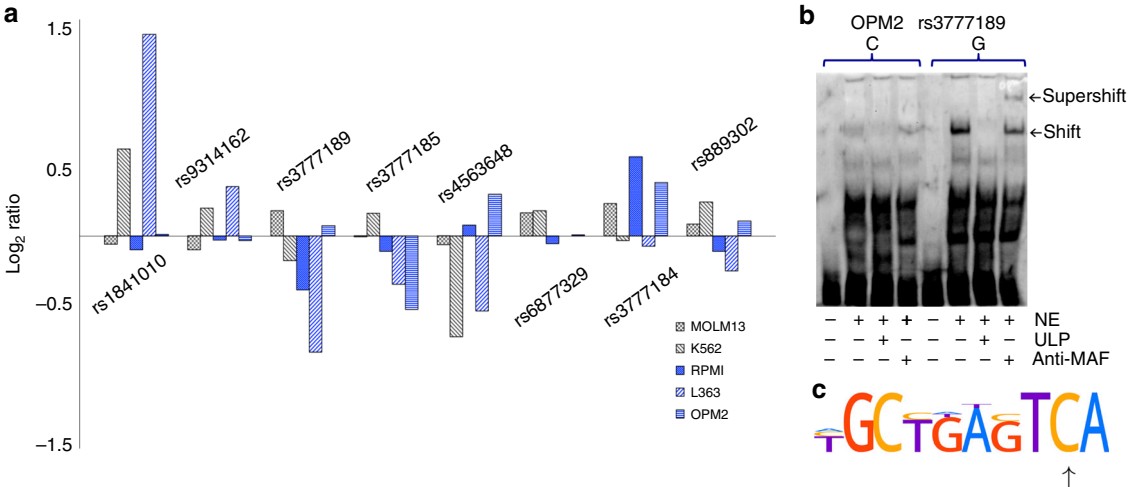

**Fig. 3** Identification of causal variants. **a** We evaluated eight variants located in regulatory regions of *ELL2* (Fig. 2c and Supplementary Table 1) using luciferase assays in RPMI-8226, OPM2, and L363 plasmocytoma lines, and in K562 and MOLM-13 cells that represent other hematologic lineages. Consistent with the effect on *ELL2* expression in plasma cells but not in peripheral blood, three risk variants (rs3777189-C, rs3777185-C, and rs4563648-G) yielded reduced activity in the plasma cell lines but not in the control cell lines, whereas the remaining variants showed opposite or inconsistent effects. Plotted values represent log$_2$-ratios of luciferase activities for the risk alleles over their corresponding protective alleles (median over tri or quadruplicates). **b** To identify the responsible transcription factors, we carried out sequence analyses and electrophoretic mobility shift assays (EMSA) (Supplementary Figs. 5 and 6). Shown here is EMSA with nuclear extract (NE) from OPM2 cells (lanes 2–4 and 6–8), and probes representing genomic sequence at rs3777189 with the risk/low-expressing allele (C) or the protective/high-expressing allele (G) in the center (lanes 1–4 and 5–8, respectively). The G allele showed an allele-specific shift (lane 6) that was outcompeted with unlabeled probe (ULP; lane 7) and supershifted with MAFF/G/K antibody (lane 8). Similar results were seen with L363 cells (Supplementary Fig. 5). **c** Sequence analysis predicted loss of a MAFF/G/K motif (Supplementary Table 4). Shown here is the MAFK motif from the HOCOMOCO-10 database. Arrow indicates G changed to C by the rs3777189-C risk variant

We screened these three variants for gain or loss of transcription factor-binding motifs. We identified numerous candidate factors, about 20 of which are expressed in MM plasma cells (Supplementary Tables 4 and 5). Electrophoretic mobility shift assays (EMSAs) with L363 and OPM2 nuclear extracts revealed allele-dependent binding of nuclear proteins for rs3777189 and rs3777185, but not for rs4563648 (Supplementary Fig. 4).

To search for differentially bound nuclear proteins, we carried out EMSA assays with antibodies against factors predicted to gain or lose a binding site at rs3777189 or rs3777185. We observed supershift with antibody against the MAFF/G/K transcription factors with probes for the protective/high-expressing allele rs3777189-G, but not with probes for the risk/low-expressing allele rs3777189-C (Fig. 3b, c and Supplementary Fig. 5). Moreover, *ELL2* expression correlated with *MAFK* and *MAFG* expression (Supplementary Table 5), and rs3777189 maps to an annotated MAFK ChIP-seq peak in lymphoid cells (Supplementary Table 1). The MAF protein family (MAF, MAFA, MAFF, MAFG, and MAFK) are paralogous basic leucine zipper (bZIP)-type transcription factors that form homo and heterodimers both with each other and certain other bZIP transcription factors (e.g., BACH1)[37–39]. MAFF/G/K are thought to be functionally redundant, and have similar binding motifs (Supplementary Table 4). Our results indicate that rs3777189-C leads to loss of a binding site for at least one of MAFF/K/G, and thereby reduced transcriptional drive. No additional supershifts were identified for rs3777189 or rs3777185 (Supplementary Fig. 6).

**The *ELL2* MM risk allele upregulates ribosomal genes**. ELL2 is a key component of the SEC. Accordingly, variation in *ELL2* expression could influence gene expression in a broader sense, either through modulation of RNA polymerase II or through cellular responses to altered protein synthesis. Consistent with this notion, mouse studies have shown that *Ell2* influences Ig heavy chain exon usage, and the processing of a large percentage of transcripts in plasma cells[22,23,40].

To gain insight into the downstream effect of variation in ELL2 function, we first calculated the correlation between *ELL2* and other genes expressed in MM plasma cells in the Swedish-Norwegian mRNA-sequencing data, which had high sequence coverage (about 100 million reads per sample) and allow accurate, linear estimation of transcript levels. Here, *ELL2* showed a significant correlation with a large set of genes, including 4890 genes with <5% false discovery rate (Supplementary Data 1). Interestingly, gene set enrichment analysis showed an over-representation of positive correlations among multiple gene sets related to ribosomal biogenesis and function (Supplementary Table 6), including a set of 80 genes encoding the proteins of the large and small ribosomal subunits (ribosomal protein coding genes, RPGs) and a set of seven genes encoding other members of the SEC (Fig. 4a)[41]. These results are consistent with co-regulation of cellular components required for high-rate protein synthesis, and the role of ELL2 in driving the production of secreted Ig.

Next, we correlated the *ELL2* MM risk allele with the expression of other genes in the mRNA-sequencing dataset. Compared to the signature obtained by correlating with *ELL2*

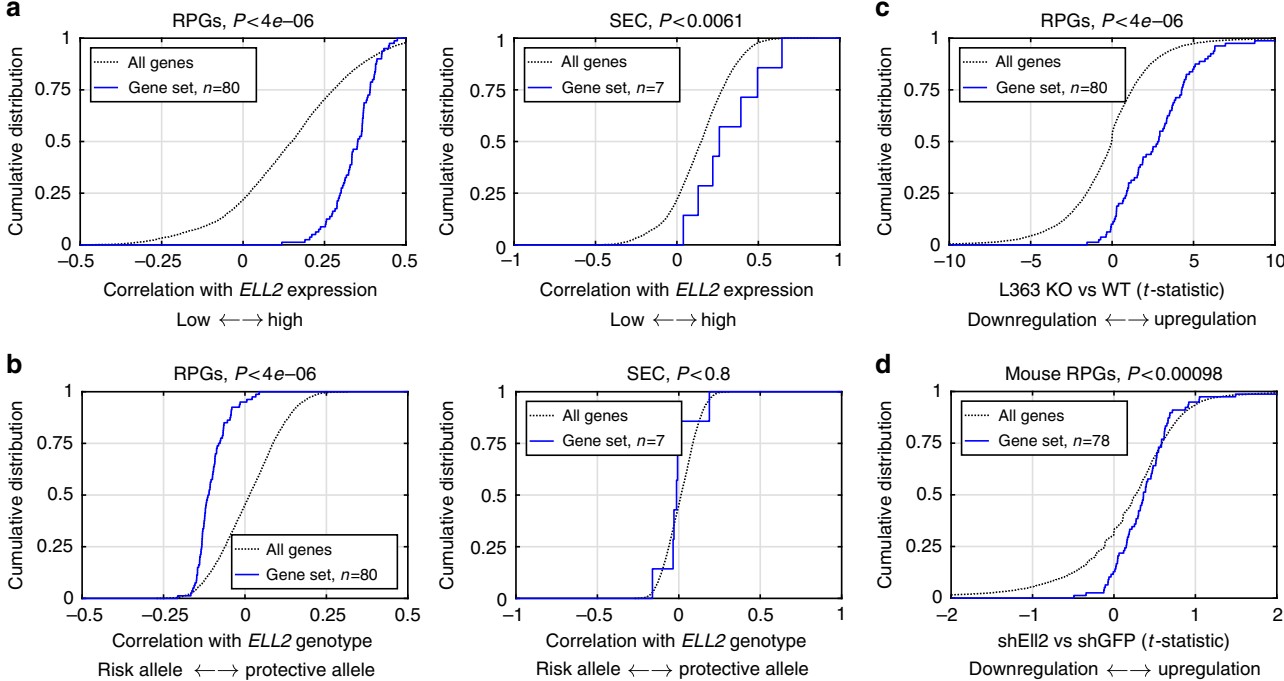

**Fig. 4** The *ELL2* MM risk allele increases ribosomal gene expression. **a** To explore the downstream effects of reduced ELL2 function, we first calculated the correlation between *ELL2* and other genes in the Swedish-Norwegian mRNA-sequencing data. Here, *ELL2* showed significant correlation with a large set of genes. Enrichment analysis revealed an over-representation of positive correlations among multiple gene sets related to ribosomes biogenesis and function, including ribosomal protein coding genes (RPGs) and the SEC (see also Supplementary Table 6). **b** Enrichment analysis of correlation between the *ELL2* MM risk allele and gene expression in the same dataset identified RPGs and other gene sets related to ribosomes. This enrichment was in the direction of the *ELL2* risk allele, which confers lower *ELL2* expression (see also Supplementary Table 7). **c** Similarly, analysis of *ELL2* CRISPR-Cas9 knockout (KO) L363 cells showed an upregulation of RPGs and other gene sets related to ribosome biogenesis and function (see also Supplementary Tables 8 and 9), i.e., effects in the same direction as the *ELL2* MM risk allele. **d** Finally, similar changes were seen in mouse MPC1 plasmocytoma cells treated with shRNA against either *Ell2* vs GFP. These data support that, in addition to the effect on *ELL2* itself, the *ELL2* MM risk allele confers additional changes in gene expression, including an increased expression of genes involved in ribosomal biogenesis, possibly as a compensatory reaction

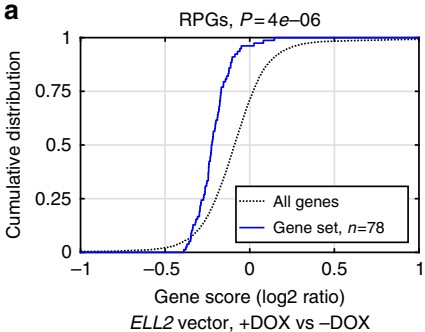
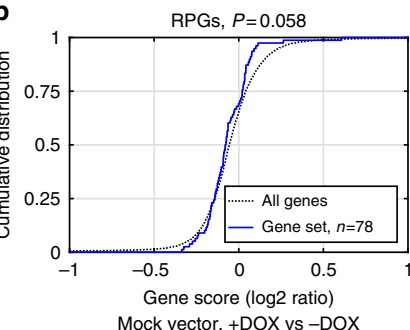

**Fig. 5** Reconstitution of *ELL2* expression. For further validation of the effect of *ELL2* knockout on RPG expression, we transduced CRISPR-resistant, doxycycline-inducible *ELL2* or mock vector into our L363-*ELL2*-KO cells. Following culture with or without doxycycline (DOX), the cells were gene expression-profiled using mRNA sequencing: **a** Comparing the gene expression profiles of *ELL2*-transduced cells cultured with ($n = 3$) vs without DOX ($n = 4$), we observed an enrichment of negative gene scores (i.e., downregulation) of ribosomal gene sets, consistent with a rescue effect; **b** no similar enrichment was seen with mock-transduced control cells ($n = 4$ samples with vs 4 without DOX). These data further support a cause–effect relationship, and that the results in Fig. 4b are not due to CRISPR-Cas9 off-target effects

expression, this signature was weaker (Supplementary Data 2), which is expected as the *ELL2* MM risk allele only explains a part of the variance in *ELL2* expression. Yet, gene set enrichment analysis again identified gene sets related to ribosome biogenesis and function (Fig. 4b and Supplementary Table 7). Unexpectedly, the detected enrichment was in the direction of the *ELL2* MM risk allele, which confers lower *ELL2* expression.

To understand whether the association with ribosomal gene expression reflects a cause–effect relationship, we knocked out *ELL2* in L363 cells using lentiviral CRISPR-Cas9 (Supplementary Fig. 7), and analyzed knockout and wild-type cells by mRNA sequencing. Strikingly, L363-*ELL2*-KO cells showed a significant enrichment of increased expression for RPGs (Fig. 4c) and other gene sets related to ribosome biogenesis and function (Supplementary Tables 8 and 9). We also observed a similar trend in pre-existing mRNA-sequencing data from mouse plasmocytoma cells treated with short hairpin RNA against Ell2 or GFP (Fig. 4d). To exclude off-target effects of CRISPR-Cas9 editing or lentivirus integration, we carried out rescue experiments where *ELL2* expression was reconstituted in the L363-*ELL2*-KO cells. For this, we generated a vector containing CRISPR-resistant *ELL2* controlled by a doxycycline-inducible promoter (Supplementary Fig. 8). *ELL2*- and mock-transduced L363-KO cells were cultured with and without doxycycline, and analyzed with mRNA sequencing. Consistent with a rescue effect, we observed doxycycline-dependent downregulation of ribosomal genes in *ELL2*-transfected cells, but not in mock-transfected cells (Fig. 5). These data support that decreased *ELL2* expression/function increases ribosomal biogenesis, possibly as a compensatory reaction in response to reduced protein synthesis.

## Discussion

*ELL2* has been associated with MM and several other phenotypes. It has been postulated that the MM risk allele has a negative effect on *ELL2* function, yet the reason for this has been unclear. We show that the MM risk allele lowers *ELL2* expression in plasma cells, providing an explanation for the hypomorphic effect. Further, we identify three risk variants that map to regulatory regions and yield decreased transcriptional activity in plasmocytoma cell lines. Two of these (rs3777185 and rs3777189) exhibit altered binding of nuclear proteins, and rs3777189-C diminishes binding of MAFF/G/K family transcription factors. In addition, we identify increased expression of ribosomal genes as a downstream effect.

Our data are consistent with a working model where the expression of *ELL2* is co-regulated with the expression of

ribosomal gene sets to allow production of secreted Ig in a coordinated manner. The MM risk allele confers lower *ELL2* expression, which makes the production of secreted Ig less efficient[13,19,21–23]. Hypothetically, plasma cells sense this and try to compensate by increasing the drive on Ig synthesis, which leads to relative upregulation of gene sets related to ribosome biogenesis and function. Such a model would explain the co-occurrence of the positive correlation between *ELL2* and ribosomal gene sets, and the negative correlation between the *ELL2* MM risk allele and ribosomal gene sets.

Regarding limitations, our study is based on plasma cells from MM patients. While it seems likely that our findings extend to normal plasma cells, it remains verify this using samples from healthy individuals. However, this is hard to do in practice as it would require isolation of CD138[+] cells from bone marrow samples from a large number of healthy volunteers. Moreover, this isolation would need to be done by fluorescence-activated flow cytometry, instead of magnetic-bead sorting, as plasma cells are rare (<1% of nucleated cells) in samples from healthy individuals. It would also be interesting to test whether our findings extend to patients with MGUS or smoldering MM. Further, while complete testing of all the linked variants that tag the *ELL2* MM risk allele would be desirable, we focused on variants in regulatory regions for practical reasons. Similarly, our data do not exclude an effect of the missense variant rs3815768 on top of the reduced expression, and we have not been able to look for effects at the protein level due to lack of material. Finally, it would be interesting to look for effects on *ELL2* and ribosomal gene sets in salivary gland samples, in light of the recently reported association with salivary gland cancer[32].

An intriguing question is how the *ELL2* risk allele promotes MM development. Hypothetically, one possibility is that the lower Ig levels could lead to slower antigen clearance and stimulation of the B-cell system for longer periods of time, and thereby a higher risk of malignant transformation. Alternatively, it is conceivable that altered ribosome function could promote MM development owing to the connection between altered ribosome biogenesis and perturbation of oncogenic pathways (c.f., refs. [42–45] and references therein).

## Methods

**Study populations**. To generate the Swedish-Norwegian gene expression dataset, we used CD138[+] cells isolated from 185 bone marrow samples obtained at diagnosis from MM patients. For 158 samples, we also obtained matching DNA from peripheral blood (Swedish National Myeloma Biobank, Lund, Sweden and Norwegian Biobank for Myeloma, Trondheim, Norway). Finally, to look for effects of *ELL2* expression in peripheral blood, we used mRNA expression data for 2515

Icelandic samples (deCODE Genetics, Reykjavik; unpublished). The sample collection was done subject to informed consent and ethical approval (Lund University Ethical Review Board, 2013/54; Icelandic Data Protection Authority, 2001010157; and National Bioethics Committee 01/015), and in accordance with the principles of the Declaration of Helsinki.

For validation, we used three sets of pre-existing gene expression profiles of CD138+ plasma cells isolated from MM patients from Germany, UK, and USA[29]. The German sample set consists of 658 MM patients from the Heidelberg University Clinic and the German-speaking Myeloma Multicenter Group[29]. The British sample set comprises 183 MM patients enrolled in the UK Medical Research Council Myeloma IX trial[29]. The US sample set comprises 604 samples from newly diagnosed patients treated at the UAMS Myeloma Institute for Research and Therapy[12]. The three validation datasets were generated using Affymetrix U133 2.0 plus microarrays and custom chip definition file ("BrainArray"; http://brainarray.mhri.med.umich.edu/Brainarray/Database/CustomCDF).

**Gene expression profiling of Swedish-Norwegian samples**. For the Swedish-Norwegian samples, total RNA was purified from immune-magnetically isolated CD138+ cells using standard methods (Macherey Nagel NucleoSpin® RNA #740955.10 or QIAamp RNA blood #52304). Icelandic blood samples were collected in PAXgene tubes (PreAnalytix, Switzerland; cat no. #762165) and RNA was isolated using the PAXgene 96 Blood RNA or the Paxgene Blood RNA Kit (PreAnalytix; cat nos. #762331 or #762174). The RNA integrity (RIN) was assessed using the BioAnalyzer (Agilent, Santa Clara, CA, USA) or LabChip GX (PerkinElmer, Waltham, MA, USA) instruments. Indexed sequencing libraries were prepared using the TruSeq RNA sample preparation v2 kit in 96-well format (Illumina, San Diego, CA, USA). Between 0.1 and 1 µg of total RNA was used for poly-A mRNA capture using oligo-dT attached magnetic beads. Complementary DNA synthesis was done using SuperScript II and random hexamer priming (ThermoFisher, Waltham, MA, USA). End-repair, 3′-adenylation, ligation of indexed adaptors and PCR amplification was performed according to Illumina protocols. Quantity and quality of each sequencing library was assessed using the LabChip GX, followed by standard dilutions and sample/plate storage at −20 °C. Further quality assessment was performed by doing pool sequencing (≤24 samples/pool) on a MiSeq instrument in order to optimize cluster densities and assess insert size, sample diversity, and so on. Primary processing and base calling was done using HCS1.3.8–1.4.8 and RTA1.10.36–1.12.4.2 analysis packages. Demultiplexing and generation of FASTQ files was performed using scripts from Illumina (bcl2fastq v.1.8). Sequence alignment and fragment counts was done with TopHat2 and HTSeq-count, respectively[46,47]. The plasma cell gene expression data will be deposited in the NCBI Gene Expression Omnibus (GEO) database when the manuscript is accepted. The German, UK, and US gene expression datasets were generated in previous studies using Affymetrix U133A 2.0 plus arrays with a custom chip definition file (v.17)[1,2].

**Genotyping**. The Swedish-Norwegian sample set was genotyped at two levels: first, all samples ($n = 185$) were genotyped for the ELL2 MM risk allele using the coding variant rs3815768, which could be robustly typed manually from the RNA-sequencing data using Integrative Genomics Viewer (Supplementary Fig. 1a). In addition, a subset of the Swedish-Norwegian samples was genotyped on Illumina Human OmniExpress microarrays ($n = 158$). To increase the genomic resolution, these data were haplotype-phased using SHAPEIT2 (v2.790)[48] and imputed by IMPUTE2 (v2.3.2)[49] with the 1000 Genomes Phase 3 compendium reference data (October 2014 release)[27]. The German, UK, and US myeloma sample sets were genotyped previously on Illumina Human OmniExpress-12 v.1.0 arrays[12,29] and imputed using the UK10K compendium[14,15,50]. For the Icelandic blood samples, genotypes were obtained by imputing variants identified by whole-genome sequencing of 8453 Icelanders into 150,656 chip genotyped individuals using long-range phasing based imputation[51,52]. Probabilities of genotypes were also predicted for 294,212 first and second-degree relatives of chip-typed individuals[53]. A description of the alignment to the reference genome, genotype calling, and imputation and haplotype phasing is given in a recent publication[54].

**Association testing**. In the Swedish-Norwegian sample set, test of association between the ELL2 risk variants and expression values generated from the MM plasma cell mRNA-sequencing data was done at the exon level, in order to allow detection of exon-specific effects and to avoid signal dilution due to alignment bias caused by coding variants (Supplementary Fig. 1b). For association testing, we used Pearson correlation as implemented in R (v.3.3)[55]. Effect sizes (beta, $\beta$) and standard errors (SE) of eQTLs were calculated using R (v.3.3). The coefficient of linkage disequilibrium (D′) and r-squared ($r^2$) were calculated using the Central European part of the 1000 Genomes compendium as available via HaploReg 4.1. To estimate risk allele ratios in rs3815768-CT heterozygotes, we counted the two allelic sequences ([C/T]AGCATTCTGAGACGGATTTAGTTTTC, representing the site of rs3815768) in the raw RNA-sequencing reads using BBTools (http://jgi.doe.gov/data-and-tools/bbtools). Exact matches of the variant sequence and its complement were counted. In the German, UK, and US sample sets, the association was done using MatrixEQTL under a linear model[12,29]. In the Icelandic mRNA-

sequencing dataset, we used generalized linear regression to test for association on rank-transformed expression estimates. To account for family structure, an estimate of the inverted kinship matrix was incorporated into the test[52]. Effect sizes (beta, $\beta$) and SE of eQTLs were calculated using R (v.2.8). Meta-analysis of P values for eQTL associations was performed using the Fisher's inverse $\chi^2$ test in MATLAB.

**Chromatin immunoprecipitation sequencing**. L363 cells were cross-linked with 1% paraformaldehyde (ThermoFisher, #28908) at 37 °C in water bath for 11 min. Shearing and immunoprecipitation was done according to manufacturer's instructions (Millipore, #17-10085). The DNA was sonicated between 200–400 bp fragment length on Biorupture Pico Sonication System (Diagenode) at 4 °C for 30 s/30 s and 13 cycles. To pull down fragments, we used 1–10 µg of H3K4me3 (Millipore, #04-745) or isotype control antibodies (normal rabbit IgG, #sc-2027, Santa Cruz Biotechnology). Fragments were de-cross-linked and purified using ChIP clean and concentrate kit (Zymogen, #D5205). Concentration was measured using Qubit 2.0 fluorometer. The ChIP-Seq library was prepared using ThruPLEX DNA-seq Kit (RUBICON GENOMICS, #R400406). Following amplification, samples were run on bioanalyzer to verify amplification and fragment size. The library was purified using AMPure XP protocol described in ThruPLEX DNA-seq Kit instruction manual. The library was diluted with nuclease-free water to 2 nM concentration. Dual-indexed libraries were sequenced on Illumina HiSeq 2500 sequencer using the TruSeq v4 cluster and SBS sequencing kits, respectively (paired-end; 2 × 125 cycles). Demultiplexing and generation of FASTQ files was performed using scripts from Illumina (bcl2fastq v.1.8). FastQC (v0.11.5)[56] was used to assess read quality, GC content, the presence of adaptors, over-represented k-mers and duplicated reads. Bases with low quality score were removed using Trimmomatic program (v.0.36)[57]. Trimmed reads were aligned using Bowtie2 (v.2.3.0)[58].

**Luciferase assays**. Ten double-stranded nucleotide sequences of 120 bp each, including with KpnI and BglII restriction sites at terminal ends, were commercially synthesized (Integrated DNA Technologies, USA). The sequences correspond to rs1841010, rs9314162, rs3777189, rs3777185, rs6877329, rs3777184, rs889302, and rs4563648 (Supplementary Table 3). Sequences were directionally cloned into a pGL3-Basic plasmid (Promega) upstream of a luciferase reporter gene[59]. Sanger sequencing confirmed the inserts. Renilla luciferase was used as internal transfection control. L363, OPM2, RPMI-8226, MOLM-13, and K562 cells were cultured at 37 °C and 5% $CO_2$ in RPMI 1640 medium (Gibco, Life Technologies) supplemented with 10% fetal bovine serum (Gibco). These cells were transfected with each of the ten clones using Neon system (ThermoFisher). Post 24 h transfection, cells were harvested and lysed in lysis buffer. An aliquot of 20 µl of the lysed cells was used for luciferase measurement following manufacturer's protocol (dual-luciferase reporter assay system, Promega). Measurements were performed at GLOMAX 20/20 Luminometer using Run Promega Protocol (DLR-0-INJ). Effects were quantified as $\log_2$ ratios of renilla-normalized luminiscence values for the risk alleles divided by the corresponding values for the protective alleles (median over three to seven replicates per sequence and cell line).

**Electrophoretic mobility shift assays**. For nuclear proteins and gel shifts[59,60], we used the following 25-bp double-stranded probes (variants in brackets): for rs3777189, ACAGTGCTGACT[G/C]AGCTCAAAATAC; rs3777185, CTCTGAAACTCT [G/A]CCTGAATGGCTC; rs4563648, GAAACTTTCTCA[C/T]CCTGACATTTGT. All probes were biotin-labeled at the 5′end of both strands; unlabeled specific competitor probes with identical sequences were used to test for specificity. For supershift assays with nuclear extracts from OPM2 and L363 cell lines (DSMZ, Braunschweig), we used these antibodies: BACH1 (#sc-271211, Santa Cruz Biotechnology), JunB (#3753S, Cell Signaling Technology), c-Fos (#4384S, Cell Signaling Technology), and MafF/G/K (D-12), #sc-166548, Santa Cruz Biotechnology. In essence, 1–2 µg antibody was added to the reaction mix and incubated 15 min at room temperature, before addition of probes and another 20 min incubation at room temperature. The cell line identities was confirmed by the supplier and mycoplasma was eliminated with ciprofloxacin, then confirmed negative in microbiological culture, RNA hybridization, and PCR assays (DSMZ, Braunschweig).

**Motif analysis**. To identify transcription factors whose motifs are gained or lost by sequence variants, we used PERFECTOS-APE (http://opera.autosome.ru/perfectosape) with the HOCOMOCO-10, JASPAR, HT-SELEX, Swiss Regulon and HOMER motif databases and default parameters ($P < 0.0005$ for both the reference and alternative variant; fold change >5).

**Knockout using CRISPR-Cas9**. To knock out ELL2 in L363 cells, we used CRISPR-Cas9 vectors encoding two different single-guide RNAs (sgRNAs) corresponding to DNA sequences TCTGGTAAGTCTCGAGCGCCCGG (clone #6) and TGCGGGAGGAGCAGCGCTATGGG (clone #2.3). These sequences, which were designed using the CRISPR Design tool (http://www.crispr.mit.edu-tool) and target ELL2 exon 1, were synthesized and ligated into lentiCRISPRv2 vector (AddGene, Cambridge, MA, USA; cat. no. #52961) using published protocols[61]. An aliquot of

ligated mix was transformed to JM109 competent cells. sgRNA inserts were confirmed by Sanger sequencing using standard Hu6-F primer. The lentiCRISPRv2 vector containing inserts were transfected into L363 cells by electroporation and puromycin selection. Successful knockout was verified by western blot with antibodies toward ELL2 (Santa Cruz Biotechnology, cat. no. sc-376611). For this, five million cells were collected and washed with PBS. Cells were lysed using 2× Laemmli sample (100 μl) and 2-mercaptoethanol. Samples were kept on ice and sonicated on Bioruptor-pico (Diagenode) for ten cycles at 30 s/30 s on and off. Thereafter, samples were heat denatured at 96 °C for 5 min and centrifuged at full speed for 5 min. Supernatant was transferred to another vial and loaded on gel. For protein separation and blot, we used mini-protein TGX stain free gel (Bio-Rad) and trans-blot turbo transfer pack (nitrocellulose, Bio-Rad) followed by overnight incubation with ELL2 antibody (Santa Cruz Biotechnology, #37661) and development (Bio-Rad). Membranes were re-probed with GAPDH antibody after re-blot treatment (Millipore, #2502).

**Analysis of cell line data and gene set enrichment analysis**. From wild-type and CRISPR-Cas9 *ELL2* knockout cells, we purified and sequenced mRNA using the same protocols as the primary CD138$^+$ plasma cell samples. Two replicates from wild-type cells and two replicates from each of two independent clones (clone #6 and clone #2.3) were analyzed. Differentially expressed genes were identified by comparing FPKM (fragments per kilo base of exon per million fragments mapped) values using Smyth's moderated *t*-statistic[62]. For gene set enrichment analysis, we used the RenderCat[63] tool with default parameters, Gene Ontology[64] and ABI Panther (http://panterdb.org) gene set databases, and considered genes with average FPKM >5 in the MMPC RNA-sequencing data. We also created specific gene sets comprising the ~80 genes encoding the proteins of the large and small ribosomal subunits ("RPG") and 7 genes encoding other members of the super-elongation complex ("SEC"). In addition to the L363 gene expression data, we used pre-existing gene expression profiles of shEll2- vs shGFP-treated mouse MPC1 plasmocytoma cells. These data were retrieved from the NCBI Gene Expression Omnibus Omnibus (accession no. GSE40285). The MPC1 data were analyzed using the same methods as the L363 data.

**Reconstitution of *ELL2* expression in L363-*ELL2*-knockout cells**. To reconstitute *ELL2* expression in the L363-*ELL2*-KO cells generated using CRISPR-Cas9, we inserted *ELL2* into a Tet-ON-3G doxycycline-inducible gene expression system (Clontech). To allow the construct to escape CRISPR-Cas9 editing, we changed the sixteenth *ELL2* codon from GGG to GGC, both coding for glycine. The new codon change eliminates the PAM sequence of the sgRNA that was used to generate the L363-*ELL2*-KO cells. The coding mRNA transcript (based on NM_012081.5, 351-2273) was synthesized as gBlocks Gene Fragments from IDT. The gene fragment was cloned in pTRE3G inducible vector. The L363-*ELL2*-KO (clone #2.3) were electroporated with pTRE3G-*ELL2* and pTRE3G-EF1α (Clontech) at a ratio of 4:1 using the NEON system (Thermo-Fisher Scientific). For mock/control transfection, we used Empty pTRE3G- and pTRE3G-EF1α (Clontech). The electroporated cells were cultured with or without doxycycline (200 ng/ml) for 24 h. RNA was prepared using the RNeasy mini kit (Qiagen), quality-assessed using Nanodrop and Bioanalyzer (Agilent), and sequenced using 2 × 75-bp Illumina mRNA sequencing at the Centre for Translational Genomics facility (Lund University), yielding about 36 million paired-end reads per sample on average. Sequences were aligned to hg38 reference genome using TopHat, and expression (FPKM) values were quantified using CuffLinks[47]. Successful induction of *ELL2* expression was confirmed by western blot, and by the presence of reads containing the new glycine codon in the RNA-sequencing data in the doxycycline-treated samples. Differential gene expression was quantified using log$_2$ ratios, and enrichment analysis was done with RenderCat[63].

**Data availability**. The RNA-sequencing data for wild-type and knockout L363 cells, and for L363 cells from the *ELL2* addback experiments are available via the NCBI Gene Expression Omnibus (accession nos. GSE111199 and GSE111210). The eQTL data for MM plasma cells and ChIP-seq data for L363 cells are available from the authors on a collaborative basis.

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

## Acknowledgements

This work was supported by the Swedish Foundation for Strategic Research (KF10-0009), the Knut and Alice Wallenberg Foundation (2012.0193), the Swedish Research Council (2012-1753), Cancerfonden (2017/265), ALF grants from Region Skåne, the Medical Faculty at Lund University, the Swedish Society of Medicine, the Crafoord Foundation, the Arne and Ingabritt Lundberg Foundation (2017-0055), the Borås Cancer Foundation, Deutsche Krebshilfe, the Multiple Myeloma Research Foundation, and the German Ministry of Education and Science (Cliommics, 01ZX1309B). We thank S. Jónsson, Ó. Magnússon, G. Halldórsson, D. Gudbjartsson, I. Jonsdóttir, U. Thorsteinsdóttir, and K. Stefánsson for their kind assistance with mRNA sequencing, providing access to gene expression data for peripheral blood from Icelanders, and insightful comments on the manuscript. We thank R. Houlston for providing access to gene expression data for plasma cells from patients from the United Kingdom. We thank A. Collin and M. Soller and the Swedish National Myeloma Biobank for their assistance with the sample collection. We are indebted to the patients who participated in the project.

## Author contributions

M.A., A-K.W., M.H., U.G. and B.N. designed research. M.A., A-K.W., A.N., N.W. and B. N. analyzed data. I.T., M.H., H.N., A.W., O.W.S., G.M., H.G., K.H., U.-H.M. and T.M. contributed samples or data. R.A., B.S. and E.J. carried out experiments.
M.A., A-K.W., N.W. and B.N. drafted the manuscript. All authors contributed to the final manuscript.

## Additional information

**Competing interests:** The authors declare no competing interests.

