## [Peer Review File · Nature Communications]

Reviewers' comments:

Reviewer #1 (Remarks to the Author):

This is an interesting study that has followed up an association with multiple myeloma. The authors have used an impressive number of samples and they were able to show that an eQTL for ELL is operating in CD138+ plasma cells of a large cohort of MM patients where the risk allele leads to downregulation of ELL. EMSAs and luciferase assays revealed reduced binding of MAFF/G/K transcription factors to one of the risk SNP alleles (rs3777189-C). Moreover the authors observed a correlation with the ELL risk allele (which is associated with down-regulation of ELL) and upregulation of RPGs which is of interest with respect to disease pathogenesis. I have only minor comments:

Page 7. The authors identify rs3815768 as an eQTL, and show altered binding of rs3777189. However, they then talk about rs1423269 without putting it into context. It would also be helpful if they could provide the r^2 values between all the SNPs discussed in this article in a figure or table.

The EMSAs shown in supplementary Figure 4 are confusing (for example, different proteins appear to be binding to the two different alleles of rs3777189 in OMP2 cells, but this is not discussed) and it is not clear why the addition of competitor increases binding to the T allele of rs3777185.

Page 5, line 4:

And genotyped... "for" (not "with").

Differences in splicing were investigated between genotype groups using replicate multivariate analysis of transcript splicing (rMATS)

Supplementary Figure 1.

First line: To genotype.... (remove "the Genotyping of")...

And correct other grammatical errors in this paragraph.

Reviewer #2 (Remarks to the Author):

I have read with interest the manuscript by Ali et al, titled "The multiple myeloma risk allele at 5q15 confers lower expression of ELL2 and increased expression of ribosomal genes in malignant plasma cells".

I have a few major and few minor points regarding the manuscript.

MAJOR:

The authors stated in the introduction that the goal of the study is to provide mechanistic insight regarding the function of 5q15 as a risk allele for multiple myeloma. However functional studies are limited. Moreover, experiments related to ELL2 KO via Crispr/Cas9 were provided as supplementary material and in fact not uploaded at the time of submission. In fact, I was not able to find supplementary figure 7 which is truly the key mechanistic figure of the paper.

Assuming that the authors achieved successful ELL2 KO (based on the western blot that was not uploaded), it would be crucial to demonstrate that adding back ELL2 in ELL2 KO cells results in decreased ribosomal biogenesis. This is particularly true as a lentiviral CRISPR was used, increasing risks of off target effects. When attempting to establish a cause-effect relationship, addback experiments are crucial.

Second, data regarding ELL2 expression and ribosomal biogenesis genes are contradictory. In fact the authors report that the ELL2 MM risk allele, which results in reduced ELL2 expression, correlates with increased expression of ribosomal genes. However, they also reports that ELL2 expression per se, also positively correlated with ribosomal biogenesis and function genes. In the CRISPR-Cas9 ELL2 KO experiments, the authors again report that loss of function of ELL2 results in increased ribosomal biogenesis genes. These data are contradictory and the authors do not offer any potential explanation for these discrepancies. Based on these data, I am not sure the authors have in fact uncovered the mechanism by which the ELL2 MM risk allele function.

MINOR:

The authors use plasma cell and myeloma interchangeably why these terms are not synonyms. This point should be addressed;

while the authors cite technical limitations in assessing the role of ELL2 in normal plasma cells, I think it would be feasible to use MGUS or smoldering myeloma cells;

all mechanistic insight on the role of ELL2 in myeloma are derived based on mRNA expression or EMSA, it would be helpful to see protein levels of both ELL2 as well as putative targets in MM;

discussion should be expanded as far as limitations and future directions;

K562 is a erythroblastic leukemia and MOLM-13 a monocytic leukemia cell line, please correct; supplementary material figure 6 and 7, as stated above, are missing.

Based on these concerns, I would recommend a major revision.

Change list

We thank the two anonymous reviewers and editorial team for their constructive comments. All points have now been addressed, with changes as follows:

Changes in response to comments from Reviewer #1

“This is an interesting study that has followed up an association with multiple myeloma. The authors have used an impressive number of samples and show that an eQTL for ELL2 is operating in CD138+ plasma cells of a large cohort of MM patients where the risk allele leads to downregulation of ELL2. EMSAs and luciferase assays revealed reduced binding of MAFF/G/K transcription factors to one of the risk SNP alleles (rs3777189-C). Moreover the authors observed a correlation with the ELL2 risk allele (which is associated with down-regulation of ELL2) and upregulation of RPGs which is of interest with respect to disease pathogenesis. I have only minor comments.”

THANKS. We thank the reviewer for his/her encouragement. We are glad to hear that our work was appreciated.

“Page 7. The authors identify rs3815768 as an eQTL, and show altered binding of rs3777189. However, they then talk about rs1423269 without putting it into context. It would also be helpful if they could provide the r^2 values between all the SNPs discussed in this article in a figure or table.”

DONE. A list of r^2 values between all the SNP discussed has now been inserted as a new **Supplementary Table 2**. The other supplementary tables have been renumbered accordingly.

“The EMSAs shown in Supplementary Figure 4 are confusing (for example, different proteins appear to be binding to the two different alleles of rs3777189 in OPM2 cells, but this is not discussed) and it is not clear why the addition of competitor increases binding to the T allele of rs3777185.”

DONE. The legend and annotation of **Supplementary Figure 4** has now been improved. The relevant EMSA bands have been indicated, and the legend has been clarified.

“Page 5, line 4: And genotyped... “for” (not “with”).”

DONE. Wording corrected (page 5).

“Page 6: Differences in splicing were investigated between genotype groups using replicate multivariate analysis of transcript splicing (rMATS).”

DONE. Sentence clarified (page 5).

“Supplementary Figure 1. First line: To genotype.... (remove “the Genotyping of”)... and correct other grammatical errors in this paragraph.”

DONE. Legend corrected.

Changes in response to comments from Reviewer #2

“I have read with interest the manuscript titled ‘The multiple myeloma risk allele at 5q15 confers lower expression of ELL2 and increased expression of ribosomal genes in malignant plasma cells’.”

THANKS. We thank the reviewer for his/her encouragement. We are happy to hear that our work was appreciated.

“Experiments related to ELL2 KO via Crispr/Cas9 were provided as supplementary material and in fact not uploaded at the time of submission. In fact, I was not able to find supplementary figure 7 which is truly the key mechanistic figure of the paper.”, “Supplementary Figure 6 and 7 are missing.”

DONE. We thank the reviewer for bringing this to our attention. The missing supplements have now been uploaded. **Supplementary Fig. 6** provides additional EMSA results. **Supplementary Fig. 7** shows that we achieved total ELL2 knockout in L363 cells.

“Assuming that the authors achieved successful ELL2 KO (based on the western blot that was not uploaded), it would be crucial to demonstrate that adding back ELL2 in ELL2 KO cells results in decreased ribosomal biogenesis.”

DONE. This is an excellent point. As requested, we have carried out addback experiments where *ELL2* expression was reconstituted in L363-*ELL2*-KO cells. For this, we created a vector containing *ELL2* controlled by a doxycycline-inducible promoter. To allow the construct to escape CRISPR-Cas9 editing, we changed the sixteenth *ELL2* codon from GGG to GGC, both coding for glycine. This change removes the PAM site of the sgRNA that was previously transduced into the L363-*ELL2*-KO cells. The L363-KO cells were transfected with doxycycline-inducible *ELL2* or mock vector were cultured for 24 hours with and without doxycycline, and analyzed by mRNA sequencing. Successful *ELL2* reconstitution was verified by Western blot (new **Supplementary Fig. 8**). Consistent with a rescue effect, gene set enrichment analysis showed downregulation of ribosomal gene sets in *ELL2*-transfected cells cultured with vs without doxycycline, but no similar effect in mock-transfected cells (new **Fig. 5**). These data support that the effect shown in **Fig. 4c** is not due to off-target effects, and further support a cause-effect relationship.

“The authors use plasma cell and myeloma interchangeably why these terms are not synonyms. This point should be addressed”

DONE. Terminology checked and clarified throughout (minor changes indicated in blue).

“While the authors cite technical limitations in assessing the role of ELL2 in normal plasma cells, I think it would be feasible to use MGUS or smoldering myeloma cells; all mechanistic insight on the role of ELL2 in myeloma are derived based on mRNA expression or EMSA, it would be helpful to see protein levels of both ELL2 as well as putative targets in MM; discussion should be expanded as far as limitations and future directions”

DONE. We agree that these would be interesting future directions, and have expanded the Discussion section as requested (page 11-12).

“Data regarding ELL2 expression and ribosomal biogenesis genes are contradictory. In fact the authors report that the ELL2 MM risk allele, which results in reduced ELL2 expression, correlates with increased expression of ribosomal genes. However, they also reports that ELL2 expression per se, also positively correlated with ribosomal biogenesis and function genes. In the CRISPR-Cas9 ELL2 KO experiments, the authors again report that loss of function of ELL2 results in increased ribosomal biogenesis genes. These data are contradictory and the authors do not offer any potential explanation for these discrepancies.”

DONE. This has now been clarified. We understand that the results may seem contradictory at first sight, however they are not. Our data fit with a working model where the expression of *ELL2* is co-regulated with the expression of ribosomal gene sets to allow production of secreted Ig in a coordinated manner (**Fig. 4a**). The MM risk allele confers lower *ELL2* expression (**Fig. 1**), which makes the production of secreted Ig less efficient. Hypothetically, plasma cells sense this and try to compensate by increasing the drive on Ig synthesis, which leads to relative upregulation of gene sets related to ribosome biogenesis and function (**Fig. 4b-d** and **Supplementary Table 9** and **10**). Such a model would explain the co-occurrence of the positive correlation between *ELL2* and ribosomal gene sets, and the negative correlation between the *ELL2* MM risk allele and ribosomal gene sets. The working model is explained at the end of Results (page 11) and in the second paragraph of Discussion (page 11-12).

“K562 is a erythroblastic leukemia and MOLM-13 a monocytic leukemia cell line, please correct”

DONE. Cell line description corrected (page 8).

Additional changes

1. Apart from the changes suggested by the referees, we had an opportunity to evaluate three more variants in luciferase assays (rs3777184, rs889302, and rs6877329). None of these showed reduced transcriptional activity in the direction of the risk allele. These data do not change the conclusions of the study, but have been added for completeness (page 8, extended **Fig. 3** and extended **Supplementary Table 3**).
2. As requested by the editor, we attach a filled-in copy of the *Nature Communications* checklist for life sciences.
3. As requested by the editor, we have checked and corrected the manuscript format so that it complies with the *Nature Communications* requirements. These changes include shortening of the title and some subheadings, rephrasing of some sections to present tense, and changes in the order of the sections.

REVIEWERS' COMMENTS:

Reviewer #1 (Remarks to the Author):

The authors have responded to my concerns. I have no other comments.

Reviewer #2 (Remarks to the Author):

I appreciate the thorough revisions of the paper by the authors. All the comments of both reviewers were properly addressed. I have no further concerns regarding the publication of this manuscript. Well done!